# BioFEG: Generate Latent Features for Biomedical Entity Linking

**Xuhui Sui**[1], **Ying Zhang**[1*], **Xiangrui Cai**[1], **Kehui Song**[1],
**Baohang Zhou**[1], **Xiaojie Yuan**[1], **Wensheng Zhang**[2]

[1] College of Computer Science, VCIP, TMCC, TBI Center, Nankai University, China
[2] Institute of Automation, Chinese Academy of Sciences, China
{suixuhui,songkehui,zhoubaohang}@dbis.nankai.edu.cn
{yingzhang,caixr,yuanxj}@nankai.edu.cn, zhangwensheng@hotmail.com

## Abstract

Biomedical entity linking is an essential task in biomedical text processing, which aims to map entity mentions in biomedical text to standard terms in a given knowledge base. However, this task is challenging due to the rarity of many biomedical entities in real-world scenarios, which leads to a lack of annotated data for them. Limited by understanding these unseen entities, traditional biomedical entity linking models suffer from multiple types of linking errors. In this paper, we propose a novel latent feature generation framework BioFEG to address these challenges. Specifically, our BioFEG leverages domain knowledge to train a generative adversarial network, which generates latent semantic features of corresponding mentions for unseen entities. Utilizing these features, we fine-tune our entity encoder to capture fine-grained coherence information of unseen entities and better understand them. This allows models to make linking decisions more accurately, particularly for ambiguous mentions involving rare entities. Extensive experiments on the two benchmark datasets demonstrate the superiority of our proposed method.

## 1 Introduction

Biomedical entity linking assigns biomedical entity mentions in texts to corresponding canonical concepts in a knowledge base, is a key task in the biomedical NLP area. This task plays a vital role in bridging unstructured text and structured biomedical knowledge bases, making it an essential component in several downstream medical-related applications, including knowledge discovery and medical diagnosis (Joseph et al., 2016), high-throughput phenotyping (Yu et al., 2015), literature searching (Zheng et al., 2015) and biomedical question answering (Lamurias and Couto, 2019).

Although many previous biomedical entity linking methods (Leaman and Lu, 2016; Phan et al.,

*Corresponding author.

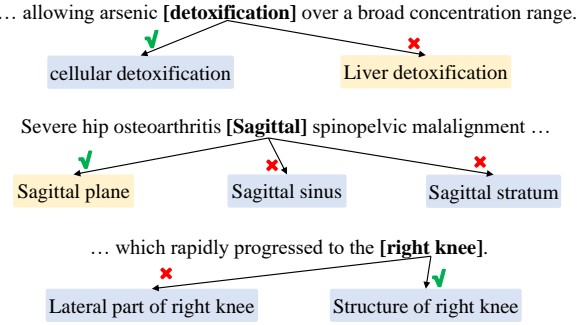

(a) Examples of three types of linking errors for biomedical entity linking. The entities in blue are unseen entities and in yellow are seen entities.

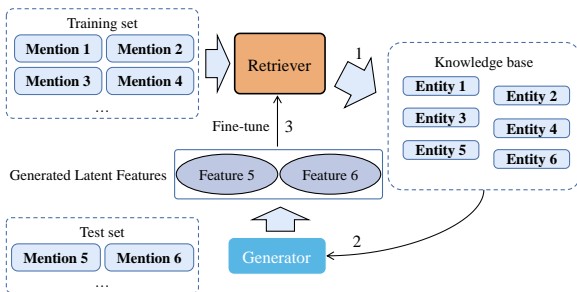

(b) Overview of our proposed BioFEG.

Figure 1: The motivation illustration of our proposed BioFEG. Linking errors may occur if unseen entities are not sufficiently understood. Our BioFEG generates latent semantic features of corresponding mentions for unseen entities to fine-tune entity encoders to capture fine-grained coherence signals of unseen entities.

2019; Sung et al., 2020; Liu et al., 2021; Yuan et al., 2022) have achieved high performance in the past few years, they often overlook entities that never appear at training, commonly known as *unseen entities*. In the biomedical field, there are many unseen entities lacking training data due to the presence of rare diseases that occur infrequently in literature. This restricts our knowledge and comprehension of unseen entities and hinders the overall performance of biomedical entity linking.

Due to the limited understanding of unseen entities, there are three types of linking errors for

biomedical entity linking, as illustrated in Figure 1a. The first type involves *linking mentions of unseen entities to wrong seen entities*, as in the top case where the mention "detoxification" may be linked to the wrong entity "Liver detoxification". The second type is *linking mentions of seen entities to wrong unseen entities*, like in the middle case where the mention "Sagittal" is confused with unseen entities "Sagittal sinus" and "Sagittal stratum". The most challenging type is the third type, which occurs when *linking mentions of unseen entities to wrong unseen entities*. As in the bottom case, the mention "right knee" can not be correctly linked without sufficiently understanding both "Lateral part of right knee" and "Structure of right knee".

Some previous works (Angell et al., 2021; Agarwal et al., 2022) have realized the challenge of unseen entities. They cluster mentions by utilizing mention-mention coreference relationships to better disambiguate mentions. However, considering the rarity of unseen entities, most clusters of them only contain few mentions. This makes it difficult to utilize mention-mention relationships of unseen entities. Thus, these methods are good at managing the second type of linking error *linking mentions of seen entities to wrong unseen entities* but still seem troubling to deal with the first and third type of linking errors.

To address these problems, we propose a novel latent **FE**ature **G**eneration framework for **Bio**medical entity linking in this paper, **BioFEG** in short. Our BioFEG is an iterative framework and each iteration contains three steps, which is shown in Figure 1b. In the first step, we train a biomedical entity linking retriever using the training set consisting of the training pairs of seen entities. To better address the second type of linking error, we utilize hard negative sampling there, which is simple but effective. In the second step, we leverage domain knowledge of seen entities to train a generative adversarial network (GAN) (Goodfellow et al., 2014) to learn to generate corresponding mention latent semantic features. In the third step, we fine-tune the entity encoder of our retriever by utilizing the generated latent features of unseen entities. This captures fine-grained coherence signals of unseen entities to simultaneously handle all three types of linking errors.

The main contributions of this paper are summarized as follows:

- For the first time, we focus on generating

pseudo data in the latent feature space for unseen entities to simultaneously deal with all three types of linking errors caused by insufficiently understanding of unseen entities.

- We propose a novel BioFEG framework, which leverages domain knowledge to generate semantically meaningful latent features for unseen entities without any labeled data and fine-tunes the entity encoder with these features to achieve higher accuracy.

- We compare our BioFEG with state-of-the-art biomedical entity linking approaches on two benchmark datasets: MedMentions (Mohan and Li, 2019) and BC5CDR (Li et al., 2016). Experimental results demonstrate the superiority of our proposed framework.

## 2 Related Work

### 2.1 Biomedical Entity Linking

Biomedical entity linking is also named as biomedical entity normalization or biomedical entity disambiguation. Diverse approaches have been proposed to explore this task in the last few years. While traditional biomedical entity linking studies (Hanisch et al., 2005; Kang et al., 2013; Cho et al., 2017) incorporate heuristic rules to normalize entities, recent state-of-the-art approaches of biomedical entity linking (Phan et al., 2019; Sung et al., 2020; Liu et al., 2021; Lai et al., 2021; Yuan et al., 2022; Zhang et al., 2022) encode mentions and entities into a common space and link mentions to the nearest entity. However, all these works ignore entities that never appear at training.

Some previous works (Angell et al., 2021; Varma et al., 2021; Agarwal et al., 2022) have focused on generalizing to unseen entities to improve the performance of biomedical entity linking. Angell et al. (2021) and Agarwal et al. (2022) utilize mention-mention coreference relationships to cluster mentions to provide another way to jointly make linking predictions. However, considering the rarity of unseen entities in the biomedical domain, most of them only have one or two mentions to group together. Thus, these methods fail to deal with the first and third types of linking errors by utilizing mention-mention relationships. Varma et al. (2021) introduces additional structural knowledge from WikiData into unseen entities to understand them better. However, the accuracy of mapping entities in biomedical knowledge bases to WikiData is only

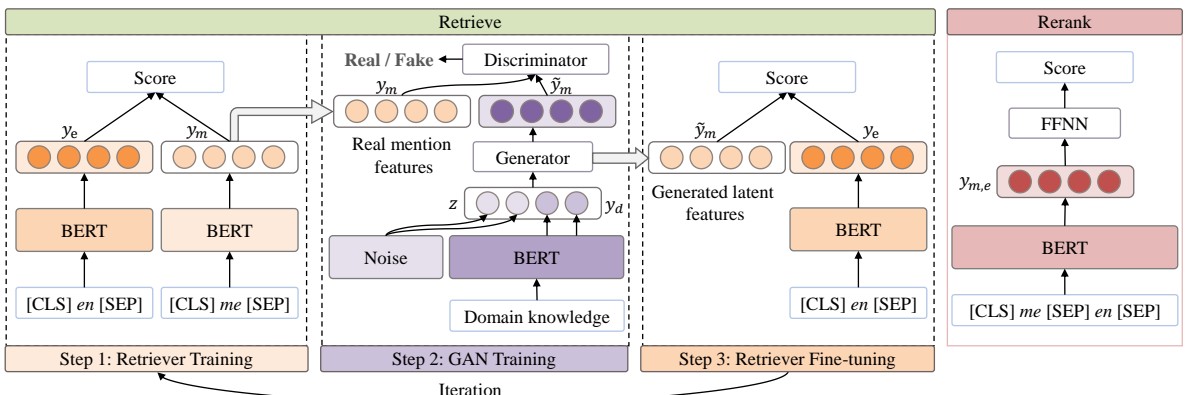

Figure 2: The overall architecture of our BioFEG framework, which contains two phases: retrieve and rerank. In our retrieve phase, it performs three steps iteratively.

80.2%. Wrong structural knowledge will have a greater negative impact.

Therefore, in this paper, we try to find another way, that is generating corresponding mention latent semantic features of unseen entities to fine-tune the entity encoder of our biomedical entity linking retriever to deal with all three types of linking errors simultaneously without introducing any cross-domain error information.

## 2.2 Zero-Shot Entity Linking

Zero-shot entity linking (Logeswaran et al., 2019) is the task that mentions need to be linked to unseen entities of new domains. Considering that we also focus on unseen entities in this paper, we briefly discuss this task there. State-of-the-art zero-shot entity linking approaches (Wu et al., 2020; Yao et al., 2020; Tang et al., 2021; Sui et al., 2022) employ a retrieve-rerank pipeline to improve the performance. Some biomedical entity linking methods (Angell et al., 2021; Varma et al., 2021; Agarwal et al., 2022) and our proposed BioFEG also follow this pipeline. However, zero-shot entity linking is a more challenging task, since all test samples are from unseen entities of new domains. In our biomedical entity linking setting, considering that the test set and training set are both from the biomedical domain and only a part of test samples are from unseen entities, there is rich domain knowledge that can be leveraged.

## 3 Methodology

### 3.1 Task Definition

Our biomedical entity linking task takes as input a biomedical text containing multiple mentions whose positions are known. Additionally, we have

a knowledge base of entities, where each entity is identified by a Concept Unique ID (CUI). Each mention with context $m_i$ is associated with its own CUI $u$. We denote the union of all entities in the knowledge base as $E = e_j{}_{j=1}^{|E|}$ where $e_j$ is a single entity and there are $|E|$ entities in the knowledge base. Our objective is to predict the gold CUI $u^*$, i.e., the target entity CUI corresponding to the mention $m_i$ as follows:

$$u^* = CUI(\text{ArgMax}_{e_j \in E} P(e_j|m_i;\theta)) \quad (1)$$

where $CUI(\cdot)$ returns the CUI of the entity $e_j$ and $\theta$ is the trainable parameters of models.

### 3.2 Overall Framework

Figure 2 shows the overall architecture of our BioFEG. Following previous works (Angell et al., 2021; Varma et al., 2021; Agarwal et al., 2022), our framework contains two phases: retrieve and rerank. Retrieve phase is to generate candidate sets for each mention from all entities of knowledge base and rerank phase is to predict the gold CUI. During the retrieve phase, it performs three steps iteratively. In the first step, we train our biomedical entity linking retriever by utilizing the training set. In the second step, we train a GAN model to learn to generate latent semantic features for corresponding mentions. In the third step, we utilize the generated latent features of unseen entities to fine-tune the entity encoder of our retriever to capture fine-grained coherence signals of unseen entities.

### 3.3 Step 1: Retriever Training

Following Varma et al. (2021), we utilize the bi-encoder architecture proposed by Wu et al. (2020) as our biomedical entity linking retriever. This

architecture contains two encoders: entity encoder and mention encoder to learn the representations of candidate entities in knowledge bases and mentions with context respectively.

The inputs of each entity and mention are respectively constructed as:

$$[CLS]\ en\ [SEP],\ [CLS]\ me\ [SEP] \qquad (2)$$

where $en = title\ [ENT]\ des$, and $me = ctxtl\ [Ms]\ mention\ [Me]\ ctxtr$. The $title$, $des$, $mention$, $ctxtl$, $ctxtr$ are the word-piece tokens of the entity title, entity description, mention, context before and after the mention respectively. [ENT] is a special token to separate the entity title and its description, [Ms] and [Me] are special tokens to tag the mention.

Both our entity encoder and mention encoder are individual BERT encoders ($\theta_{BERT_1}$, $\theta_{BERT_2}$). We feed the entity and mention inputs into them to obtain vector representations $y_e$ and $y_m$, which are the last layers of the output of the [CLS] token. The score $s(m_i, e_j)$ of each mention $m_i$ and entity $e_j$ is calculated as the inner product between corresponding representations:

$$s(m_i, e_j) = y_{m_i} \cdot y_{e_j} \qquad (3)$$

For the negative sampling, in the first iteration, our retriever is trained on in-batch negatives, which is a commonly used random negative sampling method. Within a batch, all other entities except the corresponding entity are treated as negative samples of the mention. In the following iterations, we utilize hard negative sampling to train our retriever, which chooses negative samples according to the score $s(m_i, e_j)$. The entities with higher scores except the corresponding entity are negative samples of the mention. This is a simple but effective way to handle the linking errors of *linking mentions of seen entities to wrong unseen entities*.

During the training stage, the objective of our biomedical entity linking retriever is to maximize the score of the corresponding entity of the mention with respect to negative sampling entities. Thus, for each mention-entity pair $(m_i, e_i)$, the loss function is calculated as follows:

$$\mathcal{L}_{Retr} = -s(m_i, e_i) + \log \sum_{j=1}^{N} \exp(s(m_i, e_j)) \quad (4)$$

where $e_i$ is the corresponding entity of the mention $m_i$, $N$ is the number of all sampling entities of the mention, which contains the gold entity and negative sampling entities.

## 3.4  Step 2: GAN Training

For unseen entities, the entity encoder of our retriever is never trained with corresponding mention-entity pairs due to the lack of such data. We propose to utilize GAN (Goodfellow et al., 2014) to generate pseudo data in the latent feature space for unseen entities to fine-tune the entity encoder to enhance the understanding of unseen entities.

Specifically, we utilize WGAN-GP (Gulrajani et al., 2017) to generate corresponding mention latent features based on the domain knowledge of each entity. The more domain knowledge we leverage, the more accurate the generated features will be. However, for a fair comparison, we only leverage the entity description as our domain knowledge. We utilize another BERT encoder ($\theta_{BERT_3}$) to obtain the domain knowledge representation $y_d$, which also is the output of the last hidden layer corresponding to the position of [CLS] token.

The generator combines the domain knowledge representation $y_d$ and a random Gaussian noise vector $z$ as the input while generating fake mention latent features $\tilde{y}_m$. We take mention representations in the biomedical entity linking retriever $y_m$ as our real mention latent features. The discriminator combines the domain knowledge representation $y_d$ and a latent feature vector $\overrightarrow{y}_m$ (can be a fake or real latent feature) as the input while returning a real-valued score $D(\overrightarrow{y}_m, y_d)$ to decide how realistic the $\overrightarrow{y}_m$ is. The discriminator is trained to distinguish the real feature and the generated feature, while the generator is trained to fool the discriminator. Following Gulrajani et al. (2017), the loss of our GAN model is computed as follows:

$$\mathcal{L}_G = \mathbb{E}_{\overrightarrow{x} \sim \mathbb{P}^{(\overrightarrow{y}_m, y_d)}}[D(\overrightarrow{x})] - \mathbb{E}_{\tilde{x} \sim \mathbb{P}^{(\tilde{y}_m, y_d)}}[D(\tilde{x})] +$$
$$\lambda \cdot \mathbb{E}_{\hat{x} \sim \mathbb{P}^{(\hat{y}_m, y_d)}}[(||\nabla_{\hat{x}} D(\hat{x})||_2 - 1)^2] \qquad (5)$$

where $\overrightarrow{x} \sim \mathbb{P}^{(\overrightarrow{y}_m, y_d)}$ is the joint distribution of latent features $\overrightarrow{y}_m$ and domain knowledge representations $y_d$, $\tilde{x} \sim \mathbb{P}^{(\tilde{y}_m, y_d)}$ is the joint distribution of fake latent features $\tilde{y}_m$ and $y_d$, $\hat{x} \sim \mathbb{P}^{(\hat{y}_m, y_d)}$ is the joint distribution of latent features $\hat{y}_m$ and $y_d$ while $\hat{y}_m = \epsilon \overrightarrow{y}_m + (1 - \epsilon)\tilde{y}_m$ with $\epsilon \sim \mathcal{U}(0, 1)$, and $\lambda$ is a gradient penalty coefficient. During the training stage, we only use the training set consisting of the training pairs of seen entities and minimax the loss $\min_G \max_D \mathcal{L}_G$.

## 3.5  Step 3: Retriever Fine-Tuning

After training the GAN model, we use its generator to generate corresponding mention latent seman-

tic features for unseen entities. Then we fine-tune the entity encoder ($\theta_{BERT_1}$) of our biomedical entity linking retriever with these generated latent features. The process of obtaining entity representations $y_e$ is the same as step 1. We utilize Eq. 3 to calculate the matching score $s(m_i, e_j)$ between each generated mention latent feature $\tilde{y}_{m_i}$ and obtained entity representation $y_{e_j}$. And we also choose negative samples in the hard way. For each pair $(\tilde{y}_{m_i}, e_i)$ of generated mention latent features and their corresponding unseen entities, we fine-tune the entity encoder by optimizing the loss function of Eq. 4.

### 3.6 Reranking Phase

Through the retrieve phase, we generate a candidate entity set for each mention. The goal of our reranking phase is to predict the corresponding entity from the candidate set and obtain the predicted CUI for each mention. Following previous works (Angell et al., 2021; Varma et al., 2021; Agarwal et al., 2022), we utilize the cross-encoder architecture to rerank entities. We utilize Eq. 2 to obtain the inputs of each entity and mention with context. The input of our reranking phase is the concatenation of these two inputs after removing the [CLS] token of the entity input.

We feed the input into another BERT encoder ($\theta_{BERT_4}$) to obtain the mention-candidate representation $y_{m,e}$, which is also the vector in the last hidden layer corresponding to the position of the [CLS] token. For a mention $m_i$ and one of its candidate entity $e_k$, we put the representation $y_{m_i,e_k}$ to a feed-forward neural network FFNN and obtain the matching score $s_k$ between $m_i$ and $e_k$ by using a SoftMax function:

$$\hat{s}_k = \text{FFNN}(y_{m_i,e_k}), \; s_k = \frac{\exp(\hat{s}_k)}{\sum_{j=1}^{K} \exp(\hat{s}_j)} \quad (6)$$

where $K$ is the number of candidate entities of the mention $m_i$. We utilize the cross-entropy as our loss function, which is computed as follows:

$$\mathcal{L}_{Rerank} = -l_k \log s_k - (1 - l_k) \log(1 - s_k) \quad (7)$$

where $l_k \in \{0, 1\}$, $l_k$ equals to 1 while the candidate entity $e_k$ is the corresponding gold entity of the mention $m_i$, otherwise it equals to the value 0. Finally, we can obtain the predicted gold CUI by calculating as follows:

$$u^* = CUI(\text{ArgMax}_{e_{k=1}^K} s_k) \quad (8)$$

where $CUI(\cdot)$ returns the CUI of the entity $e_k$.

| Num. of | Split | MedMentions | BC5CDR |
|---|---|---|---|
| Mentions | Train | 120K | 18K |
| | Dev | 40K | 934 |
| | Test | 40K | 10K |
| Unique Entities | Train | 19K | 2K |
| | Dev | 9K | 281 |
| | Test | 8K | 1K |
| Unseen Entities | Dev | 42.3% | 19.9% |
| | Test | 42.5% | 35.2% |

Table 1: Overall statistics of the MedMentions and BC5CDR dataset.

## 4 Experiments

### 4.1 Datasets

Following previous works (Angell et al., 2021; Varma et al., 2021), we evaluate our proposed framework BioFEG under two public biomedical entity linking datasets: MedMentions (Mohan and Li, 2019) and BC5CDR (Li et al., 2016). Table 1 shows the overall statistics of these two datasets.

MedMentions is the largest biomedical entity linking dataset, which contains 4392 abstracts from PubMed. The dataset is labeled to link to the 2017AA full-version of UMLS. Following previous works (Angell et al., 2021; Varma et al., 2021; Agarwal et al., 2022), we use the ST21PV subset of MedMentions dataset. There are a large number of entities that are unseen entities (never seen during the training stage) in the development and test set, over 42% of entities.

BC5CDR contains 1500 PubMed abstracts divided into training (500), development (500) and test (500) subsets. The mentions are linked to MESH knowledge base, which is much smaller than UMLS. In this dataset, 19.9% and 35.2% unique entities in development and test set never appear in the training set.

### 4.2 Implementation Details

In our experiments, following (Angell et al., 2021; Varma et al., 2021; Agarwal et al., 2022), all the BERT encoders we used are the BERT-base version. The maximum sequence length of words for mention with context, entity, and domain knowledge are all set to 128. Any string over the maximum length is truncated. We optimize all loss functions using AdamW (Loshchilov and Hutter, 2017). The evaluation metric is the accuracy. All scores are averaged 5 runs using different seeds.

For the retrieval phase, the total iteration number is 3. The learning rate is 1e-5 for training, and it is set to 1e-7 and 1e-9 for BC5CDR and MedMet-

| Models | MedMentions | | | BC5CDR | | |
| --- | --- | --- | --- | --- | --- | --- |
| | Overall | Seen | Unseen | Overall | Seen | Unseen |
| N-GRAM TF-IDF | 50.9 | 50.9 | 51.0 | 86.9 | 89.2 | 74.6 |
| BIOSYN (Sung et al., 2020) | 72.5 | 76.5 | 58.7 | 87.8 | 89.0 | 81.1 |
| SAPBERT (Liu et al., 2021) | 69.8 | 72.9 | 58.9 | 85.2 | 85.8 | **82.0** |
| INDEPENDENT (Logeswaran et al., 2019) | 72.8 | 75.9 | 61.9 | 90.5 | 94.0 | 73.6 |
| CLUSTERING-BASED (Angell et al., 2021) | 74.1 | 77.3 | 62.9 | 91.3 | 94.9 | 73.8 |
| DATA-INTEGRATION (Varma et al., 2021) | 74.8 | 79.7 | 57.8 | 91.9 | 94.5 | 77.5 |
| ARBORESCENCE (Agarwal et al., 2022) | 75.73 | **79.97** | 60.99 | – | – | – |
| BioFEG (Ours) | **76.68** | 79.91 | **65.42** | **93.39** | **95.57** | 81.19 |

Table 2: Experimental results on the MedMentions dataset and BC5CDR dataset. Seen and Unseen represent the sets of mentions whose corresponding entities are seen and unseen during training, respectively. We report the results of baselines according to (Angell et al., 2021; Varma et al., 2021; Agarwal et al., 2022). All scores of our BioFEG are averaged 5 runs using different random seeds. In the results, the highest values are in bold.

nions respectively during the fine-tuning stage. The number of negative samples is 64. For the GAN, we use a single-layer fully-connected network with hidden size of 768 for both the generator and discriminator. The gradient penalty coefficient $\lambda$ in Eq. 5 is set to 10. We train the discriminator 5 iterations in each generator training iteration. We train the GAN model 80 epochs with a learning rate of 5e-5. For the reranking phase, we choose the top 64 candidate entities as the candidate set for each mention. We train the reranker 5 epochs with a learning rate of 2e-5. Our experimental code is available here [1].

### 4.3 Baselines

For the quantitative evaluation of our proposed BioFEG, we utilize the following state-of-the-art biomedical entity linking methods for comparison. The n-gram tf-idf model is a traditional tf-idf based information retrieval approach. BIOSYN (Sung et al., 2020) utilizes the synonym marginalization technique to learn biomedical entity representations. SAPBERT (Liu et al., 2021) is a self-alignment pre-training approach to represent biomedical entities. INDEPENDENT (Logeswaran et al., 2019) uses the cross-encoder architecture to make linking decisions. CLUSTERING-BASED (Angell et al., 2021) is the first work to consider the problem of unseen entities in biomedical entity linking. It utilizes mention-mention coreference relationships to provide another way to jointly make linking predictions. DATA-INTEGRATION (Varma et al., 2021) introduces additional structural knowledge from WikiData into biomedical entities to enhance entity representations. ARBORESCENCE (Agarwal et al., 2022) builds min-

imum spanning arborescences over mentions and entities across documents to further improve the CLUSTERING-BASED approach.

### 4.4 Overall Performance

Table 2 shows the experimental results of our BioFEG and other baselines on the MedMentions and BC5CDR datasets. We can observe that our BioFEG outperforms all other methods on both the two datasets and achieves new state-of-the-art performance, which demonstrates the effectiveness of our BioFEG. The overall accuracy improvement on Medmentions and BC5CDR are 0.95% and 1.49%, respectively. We can also find that methods considering the unseen entity problem (INDEPENDENT, CLUSTERING-BASED, DATA-INTEGRATION, ARBORESCENCE, and our proposed BioFEG) perform better than other methods. This is consistent with our belief that the lack of understanding of unseen entities hinders the overall performance of biomedical entity linking.

In terms of the results of MedMentions, our BioFEG achieves similar performance with ARBORESCENCE on the seen set, but notably outperforms it on the unseen set. This demonstrates the superiority of our BioFEG, which generates latent semantic features of corresponding mentions for unseen entities to fine-tune the entity encoder to capture fine-grained coherence signals of unseen entities. Regarding the results of BC5CDR, our BioFEG achieves better results than all other baselines except for SAPBERT on both the seen and unseen sets. SAPBERT introduces training pairs from the UMLS knowledge base and uses metric learning to train language model. This may result in some unseen entities having been trained in these training pairs. Though SAPBERT performs well

---

[1] https://github.com/suixuhui/BioFEG

| Models | MedMentions | | | BC5CDR | | |
|---|---|---|---|---|---|---|
| | Overall | Seen | Unseen | Overall | Seen | Unseen |
| BioFEG | **76.68** | **79.91** | **65.42** | **93.39** | **95.57** | **81.19** |
| w/o generated latent features | 74.87 | 79.18 | 59.85 | 92.93 | 95.39 | 79.23 |
| w/o hard negative sampling | 74.89 | 78.83 | 61.16 | 93.02 | 95.29 | 80.33 |
| w/o reranking phase | 74.08 | 78.73 | 57.89 | 92.45 | 95.53 | 75.31 |

Table 3: Ablation study of our proposed BioFEG. We mainly evaluate the effect of three components: the fine-tuning with generated latent features, the hard negative sampling and the reranking phase.

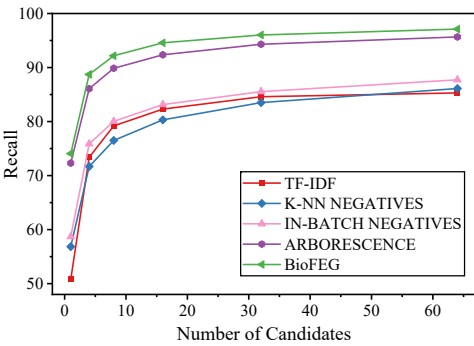

Figure 3: Retrieval results on Medmentions. The evaluation metric is recall, which measures whether the gold entity is in the top K candidates for each mention. The results of baselines come from (Agarwal et al., 2022), and the recall is micro average recall over all mentions.

on the unseen set, it appears challenging to learn good representations of seen entities. Overall, these results confirm the effectiveness of our proposed BioFEG in performing biomedical entity linking in the presence of unseen entities.

## 5 Analysis

### 5.1 Ablation Studies

To gain a better understanding of our proposed BioFEG, we conduct a series of ablation studies, as presented in Table 3. We can find that our BioFEG w/o generated latent features and BioFEG w/o hard negative sampling both result in a decrease in overall accuracy, confirming the effectiveness of these two key components. The performance of BioFEG w/o generated latent features drops more on the unseen set, while the performance of BioFEG w/o hard negative sampling drops more on the seen set. Fine-tuning with our generated latent features helps our entity encoder understand unseen entities better. Our hard negative sampling finds the most informative training samples for efficient training. It avoids easy training samples dominating the gradients to harm the learning process, which provides more fine-grained coherence information for seen entities. We also observe that our BioFEG signifi-

| Models | Type 1 | Type 2 | Type 3 | Total |
|---|---|---|---|---|
| Retrieve-Rerank | 1691 | 1203 | 1084 | 3978 |
| ARBORESCENCE | 1717 | **916** | 1070 | 3703 |
| BioFEG (Ours) | **1332** | 1089 | **1029** | **3450** |

Table 4: The number of three types of linking errors on the MedMentions test set. Type 1: *linking mentions of unseen entities to wrong seen entities*, Type 2: *linking mentions of seen entities to wrong unseen entities*, and Type 3: *linking mentions of unseen entities to wrong unseen entities*.

cantly outperforms BioFEG w/o reranking phase, emphasizing the necessity of the reranking phase after retrieving entities. The deep cross-attention between mentions and candidate entities produces more consistent gains.

### 5.2 Retrieval Performance

We report the retrieval results on MedMentions in Figure 3. We can find that our proposed BioFEG consistently outperforms all baseline approaches for all K values, which demonstrates the effectiveness of our proposed BioFEG in the biomedical entity linking retrieval phase. For recall@1, or directly linking at the retrieval phase, our BioFEG improves over ARBORESCENCE 1.77%. As the number of candidates increases, there is a tendency for all models to saturate. Despite this, we still find that our proposed BioFEG outperforms the ARBORESCENCE for 1.45% at recall@64.

### 5.3 Handling Three Types of Linking Errors

As noted in our stated contributions, our BioFEG possesses the ability to tackle three types of linking errors caused by insufficiently understanding of unseen entities. Thus, we conducted an analysis to compare the number of the three types of linking errors among BioFEG and two baselines. The ARBORESCENCE represents the current state-of-the-art, while the Retrieve-Rerank serves as the base model for both ARBORESCENCE and our BioFEG, utilizing a bi-encoder in the retrieval

| Error Type | Mention | Predicted | Annotated | Statistics |
|---|---|---|---|---|
| HO (*High Overlap*) | creatinine | creatinine | chromium | 0.6% |
| AS (*Ambiguous Substring*) | bicarbonate | sodium bicarbonate | bicarbonates | 7.7% |
| RS (*Redundant String*) | inorganic arsenic | arsenic | arsenicals | 10.5% |
| AB (*Abbreviation*) | amp | adenosine monophosphate | ampicillin | 7.5% |
| LO (*Low Overlap*) | anotia | anodontia | congenital microtia | 59.5% |
| OT (*Others*) | brain damage | brain diseases | brain injuries | 14.2% |

Table 5: Examples and statistics of each error type on the BC5CDR test set.

phase and a cross-encoder in the reranking phase. The results are shown in Table 4.

For the type 2 linking errors, ARBORESCENCE performs better than Retrieve-Rerank and our proposed BioFEG, which demonstrates the superiority of utilizing mention-mention coreference relationships to cluster mentions to provide another way to jointly make linking decisions. However, it performs worse and slightly better than Retrieve-Rerank on type 1 and type 3 linking errors, respectively. This is consistent with our claim that the methods based on mention-mention coreference relationships are good at handling the type 2 linking error but still seem troubling to deal with type 1 and type 3 linking errors.

We observe that our BioFEG significantly outperforms Retrieve-Rerank on handling all the three types of linking errors, especially on type 1 linking error. Our BioFEG utilizes the generated latent semantic features of unseen entities to fine-tune the entity encoder to capture fine-grained coherence signal of unseen entities. This is significant for biomedical entity linking models to understand unseen entities better and simultaneously handle the three types of linking errors.

### 5.4 Error Analysis

To better understand the behavior of our proposed BioFEG, we perform error analysis on the test set, with a focus on the BC5CDR dataset due to the high number of error samples in MedMentions. We manually examined all error cases and categorize them into six error types: HO (*High Overlap*), AS (*Ambiguous Substring*), RS (*Redundant String*), AB (*Abbreviation*), LO (*Low Overlap*), and OT (*Others*). The examples and statistics of each error type are shown in Table 5.

HO is an error where the mention string is identical to the predicted entity but the predicted entity is not the annotated entity. This error type may be caused by annotation errors or occur when the same mention is interpreted in different ways depending

on the context. AS is the error case whose mention string is a substring of the predicted entity, while RS is the case whose predicted entity is a substring of the mention string. AB is a case that the mention string is a form of abbreviation, which may need additional tools to replace them with the expanded form. LO denotes an error case where the mention string is very different from the surface of its gold entity. OT represents an error that does not fall into any of the above types.

In Table 5, we can observe that most errors belong to LO. This is the most challenging error type, since it requires more external knowledge and more complex reasoning to overcome the huge difference between the surface of mention string and gold entity. The available information may not be sufficient to make right linking decisions. There are also many errors are AS and RS, the mentions of them are often highly ambiguous and could refer to many different entities. We can also find 14.2% errors belong to OT, most of which are caused by the gold entity being too semantically similar to other entities, such as "brain injuries" and "brain diseases". In general, considering the limitations of annotation and available knowledge, our BioFEG has almost reached the upper bound performance of the biomedical entity linking task.

## 6 Conclusion

In this paper, we focus on rare entities in biomedical entity linking. To address the challenge of lacking training data and insufficiently understanding of unseen entities, we propose BioFEG, a novel framework to generate pseudo data in the latent feature space for unseen entities to fine-tune the entity encoder of our retriever. This captures more fine-grained coherence signals of unseen entities, allowing for simultaneously handling all the three types of linking errors caused by insufficiently understanding of unseen entities. Experimental results on two public biomedical entity linking datasets demonstrate the effectiveness of utilizing generated

latent semantic features and our proposed BioFEG achieves state-of-the-art performance.

## Limitations

Although our BioFEG has demonstrated its effectiveness in the biomedical entity linking task, there are still some limitations to be addressed in the future. The primary limitation is to generate semantically more meaningful corresponding mention latent features for unseen entities. We argue that the more domain knowledge we leverage, the more accurate the generated features will be. However, for a fair comparison with baselines, we only leverage the entity description as our domain knowledge. We will investigate to utilize more domain knowledge to generate more accurate latent features in the future work.

## Acknowledgments

This research is supported by the Chinese Scientific and Technical Innovation Project 2030 (2018AAA0102100), the National Natural Science Foundation of China (No. 62272250, 62002178, U1936206), the Natural Science Foundation of Tianjin, China (No. 22JCJQJC00150, 22JC-QNJC01580), the Fundamental Research Funds for the Central Universities (No. 63231149).

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
