# OpenReview forum: "BioFEG: Generate Latent Features for Biomedical Entity Linking"
_EMNLP/2023/Conference — EMNLP 2023 Main_

### Official Review · Reviewer_msFt · 2023-08-01

**Soundness:** 5

**Excitement:**

4: Strong: This paper deepens the understanding of some phenomenon or lowers the barriers to an existing research direction.

**Paper Topic And Main Contributions:**

The paper tackles the problem of medical entity normalization, and in particular how to improve the performance of the models on unseen entities (normalizing mentions whose entities are never seen at training time).

The proposed approach trains a mention and an entity embedder to generate meaningful embeddings on the training (seen) data. \
Domain knowledge is used to generate fake mention embeddings for unseen entities with a GAN model. \
These fake mention embeddings are then used to further finetune the entity embedder. \
An additional re-ranking step ensures maximum performance after the two previous models have retrieved the best candidate entities to normalize a mention.

The authors compare their proposed model with several strong baselines, showing improved performance on unseen entities. The ablation studies conducted also highlight the importance of the single components of the proposed architecture.


**Questions For The Authors:**

The paper is very clear and Section 5 answered all the questions I came up with while reading the first part of the paper. I don’t really have any technical questions.

- Out of curiosity, did you analyze all 3450 errors (Table 4) to get the statistics in section 5.4?
- Do you plan to release a version of your codebase?


**Reasons To Accept:**

- the proposed method is novel, interesting, and makes sense
- the method is presented in a clear and understandable way
- the ablation studies and the extra analyses answer most of the questions that a reader would have while reading the paper
- the proposed method improves the performance on unseen entity normalization compared with previous models


**Reasons To Reject:**

None that comes to mind.


**Reproducibility:**

4: Could mostly reproduce the results, but there may be some variation because of sample variance or minor variations in their interpretation of the protocol or method.

**Reviewer Confidence:**

5: Positive that my evaluation is correct. I read the paper very carefully and I am very familiar with related work.

**Typos Grammar Style And Presentation Improvements:**

- Line 55: “Figure. 1a” →”Figure 1a”
- Figure 2: minor suggestion, you could add the number of the BERT models in the image BERT_1, …, BERT_4. Although the difference is already clear thanks to the different colors, it would further tie together the text and figure, and make the figure more readable if printed in grayscale.
- Line 182: “there are rich” → “there is rich” (?)
- Line 229-235: the description of the input is clear enough, but it would be even better with an example of the input for a real entity / mention. The meaning of “title” and “description” might be unclear, I guess you are using information from UMLS, but it would be good to state it somewhere.

---

> ### Author Rebuttal · Authors · 2023-08-27
>
> Thanks for the insightful suggestions and we sincerely appreciate your positive feedback. We would like to revise all typos and address the specific questions below.
>
> 1.Analysis of 3450 errors in the MedMentions dataset
>
> We didn’t analyze the 3450 errors in the MedMentions dataset (Table 4) to get the statistics in section 5.4. We performed the statistics by entirely manual examining, and 3450 errors is a very large number. Thus, we perform the error analysis on the smaller dataset BC5CDR, which has only a few hundred errors.
>
> 2.Source code
>
> We will release the complete code for all experiments to GitHub after this paper is accepted.

---

### Official Review · Reviewer_v9F6 · 2023-08-04

**Soundness:** 4

**Excitement:**

3: Ambivalent: It has merits (e.g., it reports state-of-the-art results, the idea is nice), but there are key weaknesses (e.g., it describes incremental work), and it can significantly benefit from another round of revision. However, I won't object to accepting it if my co-reviewers champion it.

**Missing References:**

N/A

**Paper Topic And Main Contributions:**

This paper proposes a novel latent feature generation framework BioFEG to address multiple types of linking errors in biomedical entity linking. Specifically, BioFEG leverages domain knowledge to train a generative adversarial network, which generates latent semantic features of corresponding mentions for unseen entities. Utilizing these features, they fine-tune an entity encoder to capture fine-grained coherence information of unseen entities and better understand them, which allows models to make linking decisions more accurately, particularly for ambiguous mentions involving rare entities. Extensive experiments on the two benchmark datasets demonstrate the superiority of the proposed BioFEG framework.

**Questions For The Authors:**

1. What’s the key motivation and novelty of this paper?
2. Have you ever thought about the potential of your method to apply the Large Language Models (LLMs, such as GPT-4)?

**Reasons To Accept:**

1. The idea of proposing a novel latent feature generation framework BioFEG to address multiple types of linking errors in biomedical entity linking is meaningful and interesting.
2. This paper presents the design principle of the BioFEG methodology in detail, which has the potential to motivate more related works for addressing the linking errors in biomedical entity linking tasks.
3. The experimental part is rich, and a lot of studies have been done on the proposed BioFEG framework.
In general, this paper is well-organized and is a good work that can promote the development of biomedical entity linking.

**Reasons To Reject:**

1. The presentation of the paper should be improved. The expressions in the current version are not natural and elegant enough, and it would be better to improve them with the help of an English native speaker.
2. The key motivation and novelty of this paper should be discussed more.
3. The experiments are rich and diverse. But to a certain extent, this paper is lack significance and robustness analysis and it would be better to include them.
4. With the rapid development of Large Language Models (LLMs, such as GPT-4), it is necessary to discuss the potential of LLMs for this task.

**Reproducibility:**

4: Could mostly reproduce the results, but there may be some variation because of sample variance or minor variations in their interpretation of the protocol or method.

**Reviewer Confidence:**

3: Pretty sure, but there's a chance I missed something. Although I have a good feel for this area in general, I did not carefully check the paper's details, e.g., the math, experimental design, or novelty.

**Typos Grammar Style And Presentation Improvements:**

The presentation of the paper should be improved. The expressions in the current version are not natural and elegant enough, and it would be better to improve them with the help of an English native speaker.

---

> ### Author Rebuttal · Authors · 2023-08-27
>
> We thank the reviewer for the constructive suggestions and sincerely appreciate your positive feedback. We will improve the presentation to improve this paper in the revised version. We would like to address the specific questions below.
>
> 1.Key motivation and novelty
>
> In the biomedical field, there are many unseen entities lacking training data due to the presence of rare diseases that occur infrequently in literature. Limited by understanding these unseen entities, there are three types of linking errors. However, previous works are good at managing the second type of linking errors but still seem troubling to deal with the other two types of linking errors. This motivates us to find a way to simultaneously deal with all three types of linking errors to improve the performance. Thus, in this paper, we propose a novel framework to generate pseudo data in the latent feature space for unseen entities to fine-tune the entity encoder of our retriever. This captures more fine-grained coherence signals of unseen entities, allowing for simultaneously handling all the three types of linking errors.
>
> 2.Significance and robustness analysis
>
> For the significance analysis, our results over all baselines are statistically significant with p < 0.05 according to the t-test. Robustness analysis usually modifies the input to verify the robustness of the model. To the best of our knowledge, all previous biomedical entity linking works have not included the robustness analysis. In the multimodel entity linking setting, [1] designs a robustness analysis to demonstrate that their model can better utilize visual information to compensate for the textual noise. However, in our biomedical entity linking setting, visual information is not available, and the textual information in the biomedical domain is typically rigorous and complete. Therefore, we argue that designing a robustness analysis by adding textual noise in this paper is unnecessary.
>
> 3.Large language models
>
> Large language models (such as GPT-4) are pre-trained using general domain resources. However, the focus of this paper is specifically on the biomedical domain. Some previous works [2, 3] have demonstrated that large language models perform significantly worse than domain-specific language models (such as BioBERT). This is because domain-specific language models are pre-trained on vast biomedical text corpora and possess extensive domain knowledge. Hence, using LLMs is not appropriate either in this task or for our proposed method.
>
> [1] Wang et al, 2022, Multimodal Entity Linking with Gated Hierarchical Fusion and Contrastive Training, SIGIR 2022.
>
> [2] Moradi et al, 2021, GPT-3 Models are Poor Few-Shot Learners in the Biomedical Domain, CORR.
>
> [3] Gutiérrez et al, 2022, Thinking about GPT-3 In-Context Learning for Biomedical IE? Think Again, In Findings of the Association for Computational Linguistics: EMNLP 2022.

---

### Official Review · Reviewer_gxqK · 2023-08-05

**Soundness:** 5

**Excitement:**

4: Strong: This paper deepens the understanding of some phenomenon or lowers the barriers to an existing research direction.

**Paper Topic And Main Contributions:**

This paper proposes an interesting solution to biomedical entity linking where the test set contains both seen and unseen entities during training. It uses a GAN to learn latent representations for unseen entities so that the correct unseen entity has a higher chance of being captured at top k by the retriever compared to existing methods.

**Questions For The Authors:**

I wonder if the proposed method also works for other domains. It seems the framework is not specific to the bio domain at all. If the only required domain knowledge is entity descriptions, there are a lot of knowledge bases that can provide those for various domains. Is there a specific reason that you constrain your study to the bio domain?

**Reasons To Accept:**

- The problem setting is realistic: Indeed, in real-world applications, an entity linking system should better perform well on both seen and unseen entities.
- The proposed method is interesting and makes sense to me.
- The writing is clear and very easy to follow.
- The experiment section is solid and it answers almost all questions in my mind.

**Reasons To Reject:**

None. I believe this paper should be accepted.

**Reproducibility:**

4: Could mostly reproduce the results, but there may be some variation because of sample variance or minor variations in their interpretation of the protocol or method.

**Reviewer Confidence:**

3: Pretty sure, but there's a chance I missed something. Although I have a good feel for this area in general, I did not carefully check the paper's details, e.g., the math, experimental design, or novelty.

---

> ### Author Rebuttal · Authors · 2023-08-27
>
> We thank the reviewer for the insightful comments and sincerely appreciate your positive feedback. We would like to address the specific questions below.
>
> 1.Work for other domains
>
> We are confident that our proposed method can be applied effectively to other specialized domains, only if these domains meet the following three conditions: 1) a large amount of seen entity training data, 2) the existence of unseen entities that are prevalent in these domains and 3) sufficient domain knowledge to generate latent features. The bio domain meets all these conditions simultaneously, which motivated us to devise this method to generate pseudo data in the latent feature space for unseen entities to enhance overall performance. Additionally, the bio domain exhibits more advanced baselines and possesses more comprehensive datasets. Hence, this paper primarily focuses on the bio domain.

---

### Meta-Review · Area_Chair_wt6S · 2023-09-06

**Recommendation:** 5

**Metareview:**

This paper's problem setting is realistic, the proposed method is interesting and clear, and the experiments are solid and address all relevant questions.

---

### Decision · Program_Chairs · 2023-10-07

**Decision:**

Accept-Main

**Comment:**

This paper's problem setting is realistic, the proposed method is interesting and clear, and the experiments are solid and address all relevant questions.